# Preparation and Characterization of a Polyetherketoneketone/Hydroxyapatite Hybrid for Dental Applications

**DOI:** 10.3390/jfb13040220

**Published:** 2022-11-05

**Authors:** Wenhsuan Lu, Conglei Li, Jian Wu, Zhongshi Ma, Yadong Zhang, Tianyi Xin, Xiaomo Liu, Si Chen

**Affiliations:** 1Department of Orthodontics, Cranial-Facial Growth and Development Center, Peking University School and Hospital of Stomatology, 22 Zhongguancun South Avenue, Haidian District, Beijing 100081, China; 2National Center of Stomatology & National Clinical Research Center for Oral Diseases & National Engineering Laboratory for Digital and Material Technology of Stomatology & Beijing Key Laboratory for Digital Stomatology & Research Center of Engineering and Technology for Computerized Dentistry Ministry of Health & NMPA Key Laboratory for Dental Materials, Beijing 100081, China; 3Advanced Materials Division, Suzhou Institute of Nano-Tech and Nano-Bionics, Chinese Academy of Sciences, Suzhou 215123, China; 4Suzhou Xiangcheng Medical Materials Science and Technology Co., Ltd., Suzhou 215123, China; 5Division of Nanomaterials, Suzhou Institute of Nano-Tech and Nano-Bionics, Chinese Academy of Sciences, Nanchang 330200, China

**Keywords:** polyetherketoneketone, hydroxyapatite, composite, hybrid, dental, orthopedic, orthodontic, high performance, mechanical compatibility

## Abstract

Here, we developed a new synthetic method for the production of a new class of polymeric inorganic hybrid biomaterial that has potential for dental implant applications and, in general, other orthopedic applications owing to its excellent mechanical properties and biomechanical compatibility. The new hybrid biomaterial is a composite consisting of polyetherketoneketone (PEKK) and hydroxyapatite (HA). This hybrid material boasts several unique features, including its high HA loading (up to 50 wt%), which is close to that of natural human bone; the homogeneous HA distribution in the PEKK matrix without phase separation; and the fact that the addition of HA has no effect on the molecular weight of PEKK. Nanoindentation analysis was used to investigate the mechanical properties of the composite, and its nano/microstructure variations were investigated through a structural model developed here. Through nanoindentation technology, the newly developed PEKK/HA hybrid biomaterial has an indentation modulus of 12.1 ± 2.5 GPa and a hardness of 0.42 ± 0.09 GPa, which are comparable with those of human bone. Overall, the new PEKK/HA biomaterial exhibits excellent biomechanical compatibility and shows great promise for application to dental and orthopedic devices.

## 1. Introduction

Polyetherketoneketone (PEKK) is a new kind of special engineering plastic with properties that make it highly suitable for medical applications [1,2]. Accordingly, it has been approved by the Food and Drug Administration for oro-maxillofacial and spinal surgery, and it is widely used in the medical field.

It is estimated by the World Health Organization that ~60–90% children and approximately 100% of adults suffer from caries disease globally. Furthermore, ~30% of people aged between 65 and 74 have no natural teeth, with the situation being worse in poor areas. Tooth loss caused by severe periodontal disease and trauma has become the main cause of oral-function decline [3,4].

The use of artificial implants is the most important method of tooth repair. Such implants can be classified as dental implants, dental bases, and dental crowns [5]. A dental implant is surgically implanted into the upper or lower alveolar bone around the missing tooth to serve as the root. Accordingly, it represents the key part of an artificial tooth as it must bear forces, fix the tooth, and combine with the bone. This makes it the most important factor influencing the healing and regeneration of alveolar bone after surgery.

A wide range of materials have been tested and used in dentistry [6], such as metals, alloys, ceramics, polymers, biological glass, and carbons. At present, the most widely used implant materials clinically are titanium and its alloys (Ti-6Al-4V) owing to their reasonably good biocompatibility and biofunctionality among readily available materials [7]. However, they still suffer certain drawbacks; for example, their elastic modulus are ~110 GPa, which is far higher than that of jaw bone (10–40 GPa) [8]. Accordingly, the stress concentration at the surface between the dental implant and the bone can lead to implant fracture and bone resorption around the implant. A study [6] showed that a significant increase in titanium ion concentration and fluid accumulation can occur adjacent to an implant site, which may inhibited osteoblast production and promote osteoclast synthesis. Although this research has not yet been clarified, this phenomenon is still worrying. Furthermore, products made with titanium and its alloys can cause complications such as titanium allergy, and the poor light transmittance of metal implants can have a negative esthetic effect [4].

Ceramics are another kind of dental implant material, and they have been studied and utilized in this field for three or four decades. Zirconium oxide [9,10,11,12,13] is the most investigated ceramic dental material, with much of the work being performed in the laboratory rather than in clinical applications. This is because of its intrinsic weaknesses such as brittleness and poor impact resistance, which mainly results from its low-temperature degradation upon aging [14,15,16]. To address these problems, some researchers have mixed iridium oxide with zirconium oxide, but these composites exhibited unsuitable in vivo properties [17,18]. As with those of metals and alloys, the elastic modulus of zirconium oxide is high (210 GPa), which causes imbalances in stress distribution and thus stress concentration [19,20].

Polyetheretherketone (PEEK) is a linear aromatic polymer compound that is widely used in the aerospace and automotive industries as well as precision instrument manufacturing owing to its excellent mechanical properties and chemical stability. In addition, PEEK is becoming increasingly used in orthopedics and dentistry owing to its biosafety, elastic modulus, and excellent surface properties [21,22]. However, compared with PEEK, PEKK has better surface chemical modification potential, is easier to sulfonate, and has better bone apatite deposition properties. As well as these improved mechanical and chemical properties, PEKK exhibits good bone fusion and antimicrobial properties [23,24].

Research into dental implants is multidiscipline, meaning that the materials used are often required to meet biological and mechanical requirements simultaneously. As a result, new biocomposite dental materials are attracting increasing research attention in the field of dental implants [25,26]. Owing to the potential applications of organic nanostructured inorganic hybrids as bioactive high performance medical implants, their design and development is one of the most active areas of research worldwide. The most common method used to make organic and inorganic composites is through mixing [27]. However, a drawback of this mixing technology is that it cannot produce composites with high inorganic loadings at the required uniformity. This is particularly true when nanoparticles or nanostructured inorganic fillers are to be dispersed in an organic polymer matrix [27]. This is because adding inorganic materials to an organic matrix is challenging due to their general chemical and physically incompatibility. Therefore, mechanical mixing processes cannot produce designed structures with nanoscale uniformities. Currently, there are no established methods for the controlled introduction of nanostructured inorganic fillers into organic polymer matrixes, so it is difficult to achieve truly nanostructured composites with high inorganic loading. This difficulty is more pronounced when scale-up and reproducibility must be considered at the manufacturing stage.

PEKK is a type of high-performance semi-crystalline thermoplastic polymer, with high strength, relatively good ductility, heat resistance, corrosion resistance, and many other great properties. Accordingly, it can be used to substitute metals and ceramics as dental implant materials [1,28,29]. Furthermore, its utility in the fields of bone implants, trauma-treatment materials, and skull reconstruction has been widely reported [1,4]. Our area of research interest is the use of PEKK to develop polymeric inorganic hybrids with elastic moduli and densities close to those of human bone, and to make PEKK bioactive and suitable for cell attachment.

There have been several studies on the mechanical properties of hydroxyapatite (HA)-whisker-reinforced PEKK scaffolds. However, their mechanical properties need to be improved if they are to be used in dental implants [30]. Synthetic methods rather than the mechanical mixing of an organic and inorganic material of opposite characteristics are required for the precise engineering of inorganic–organic hybrid biomaterials. Accordingly, the current work details a methodology to synthesize a polymeric–ceramic hybrid via a bottom-up approach, i.e., starting from a monomer and growing polymeric molecules onto nanostructured ceramic particles. In this work, PEKK as well as HA is used to synthesize such composites. The HA content of human bone is ~50–70 wt%, so the core task of the present work was to prepare hybrids with HA contents in that range. PEKK is synthesized at room temperature and normal pressure via a Friedel–Crafts reaction [31,32,33]. The unique aspect of this new synthetic strategy is that inorganic particles are employed as starting materials and PEKK are grown from monomers onto these inorganic filler particles. Specifically, HA nanoparticles are used as starting materials to make PEEK-HA hybrid composites, providing materials with very high HA loadings that also have highly uniform nanoscale morphological structures. We assume that the HA and PEKK composites synthesized in situ can improve the mechanical properties (such as indentation modulus) of the materials, so that they can be used in dental implants, as well as for other orthopedic applications.

## 2. Materials and Methods

### 2.1. Materials

Commercially available HA (rod-shaped particles; 30 nm × 100 nm) was obtained from Shanghai Hualan Chemical Technology Co., Ltd. (Shanghai, China) Diphenyl ether (>99.0%), aluminum chloride (AR, 99%), terephthaloyl dichloride (99%), and *N*-methyl-2-pyrrolidinone (99.5%) were obtained from Shanghai Aladdin Reagent Co., Ltd. (Shanghai, China) 1,2-Dichloroethane (AR) and anhydrous methanol (AR) were obtained from Shanghai Sinopharm Chemical Reagent Co., Ltd. (Shanghai, China).

### 2.2. Synthesis of PEKK/HA Composites

1,2-Dichloroethane (50 mL) and AlCl_3_ (5 g) were stirred in a three-necked flask for 30 min under nitrogen. HA (1.6 g) was then introduced to the reaction system and, after 30 min, a mixed solution of *N*-methylpyrrolidinone (1.25 mL) and 1,2-dichloroethane (5 mL) was added slowly, keeping the temperature of the reaction system at −15 °C. After 30 min, 1.5 mL diphenyl ether and 2.03 g terephthaloyl dichloride were added to the system, again maintaining the temperature at −15 °C. After 1 h, the cooling apparatus was switched off and the reaction was allowed to proceed overnight at room temperature for 20 h. Then, 400 μL diphenyl ether was added and the reaction was stirred for 120 min to terminate the polymers, and 100 mL anhydrous methanol was added and the reaction was stirred for another 120 min to stop the reaction. After washing with icy water and anhydrous methanol and separating the product from the washing solution by filtering, the product was dried in a drying oven at 80 °C for over 24 h.

### 2.3. Characterization

#### 2.3.1. Back Reflection Infrared Spectroscopy

The back reflection mode of the infrared spectrometer (Nicolet iN10; MA, USA) was used to test the samples. When testing, a small amount of HA, PEKK or PEKK/HA powders were spread on the sample table for testing. The original data of the test were drawn and analyzed with QRIGIN software.

#### 2.3.2. X-ray Photoelectron Spectroscopy

In this experiment, X-ray Photoelectron Spectroscopy (PHI 5000 VersaProbe II; Kratos; Japan) instrument is used to test the carbon and calcium elements in PEKK/HA, and the tested PEKK/HA samples were in powder form. Advantage software was used for XPS data analysis.

#### 2.3.3. Scanning Electron Microscopy and Transmission Electron Microscopy

Scanning electron microscopy (Quanta FEG 250; Hillsboro, OR, USA) along with transmission electron microscopy (Tecnai G2 F20 S-TWIN; PA, USA) were used to observe whether the HA was wrapped by the PEKK. EDS patterns of scanning electron microscopy (Quanta FEG 250; Hillsboro, OR, USA) was used to obtain information on the distribution of carbon, calcium, and phosphorus in each sample. The tested data graphs were directly used for drawing.

To prepare SEM samples, PEKK or PEKK/HA composite powders were pressed into bulk samples (1cm × 1 cm) by thermo-compression molding. Then, the bulk samples were pasted onto a conductive adhesive, and the surface of the samples were sprayed with gold (20 mA for 2 min).

To prepare TEM samples, after the reaction was stopped with anhydrous methanol, a drop of the reaction solution was pipetted to a glass bottle and diluted to 15 mL with anhydrous methanol. After ultrasonic mixing for 10 min, the solution was dropped onto a copper mesh using a capillary burette three times and then dried in an oven at 100 °C for 12 h.

To prepare EDS samples, pure PEKK and PEKK/HA composite powders were pressed into bulk samples (1 cm × 1 cm) by thermo-compression molding. Then, the bulk pure PEKK and PEKK/HA composite samples were put on conductive adhesive and adhered to the sample table, then the surface of the samples were sprayed with gold (20 mA for 2 min).

#### 2.3.4. Molecular Weight

In the research, HA was added to the reaction system to make the composites, so it was important to demonstrate that the HA does not affect the synthesis of PEKK in terms of its molecular weight. According to the Mark–Houwink formula (shown as Equation (1) below), the molecular weight of a polymer is associated with its viscosity. Because of the insolubility of HA in concentrated sulfuric acid, the HA was removed from the composites using hydrochloric acid and then the samples were dissolved in concentrated sulfuric acid and the viscosity of the solutions was determined using an Ubbelohde viscometer. The molecular weight of PEKK test samples are in powder state (**η** is viscosity of liquid; **K** is Huggins constant; **β** is Kramer constant; **M** is viscosity-average molecular weight; ***C*** is concentration of liquid; **η*_sp_*** is the specific viscosity; ηr is relative viscosity; ***α*** is a parameter related to the morphology of polymer in solution).
(1)$$\left[\eta \right]=KM^{\alpha }$$
(2)$$\frac{\eta_{sp}}{C}=\left[\eta \right]+K{\left[\eta \right]}^{2}C$$
(3)$$\frac{\eta_{sp}}{C}=\left[\eta \right]+\beta {\left[\eta \right]}^{2}C$$

**η*_sp_*** and lnηr/***C*** were plotted against ***C*** according to Equations (2) and (3), where the intercept is the intrinsic viscosity, and this can be used to calculate the viscosity average molecular weight of PEKK according to Equation (1). The original data obtained through the test were drawn and analyzed with ORIGIN software.

#### 2.3.5. Mechanical Analysis

The mechanical properties of the PEKK/HA composite are of great importance, and were measured by nanoindentation. After thermo-compression molding, a cylinder with a diameter of 13.0 mm and a height of 5.0 mm was made, and the sample was polished with sandpaper and then analyzed using an MTS Agilent G200 nanoindenter (Agilent G200; GA, USA) with the parameters: test probe, Berkovich diamond tip, tip radius 20 nm; total indenter travel, 1.5 mm; maximum indentation depth, 500 μm; maximum load (standard), 500 mN; maximum load with high load option, 10 N; load resolution, 50 nN; load frame stiffness, ≈5 × 106 N/m; useable sample area, 100 × 100 mm; maximum indentation depth, >15 μm; typical resonance frequency, 180 Hz; maximum load, 10 mN; load resolution, 1 nN; surface approach velocity, 10 nm/s; depth limit, 3000 nm; strain rate target, 0.05 per second; harmonic displacement target, 2; and Poisson’s ratio, 0.25. The original data obtained through the test were drawn and analyzed with ORIGIN software.

## 3. Results

### 3.1. Back Reflection Infrared Spectroscopy

Back reflection infrared (IR) spectroscopy was used to characterize the PEKK and the PEKK/HA composite as well as the HA, which is commercially available. The test samples of HA, PEKK, or PEKK/HA composite are in powder state. The spectra are shown in Figure 1.

The characteristic peak at ~1645.0 cm^−1^ is due to the carbonyl groups connecting two benzene rings, while the peaks around 1199.6, 1492.0, and 1589.7 cm^−1^ are due to the ether bond between two benzene rings. All these peaks can be observed for the PEKK/HA composite, indicating that it contains PEKK.

The peak at approximately 3569 cm^−1^ in the spectrum of HA is due to its hydroxyl group, whereas the peaks at 960, 1030, and 1100 cm^−1^ are the absorption peaks of the phosphate group in HA. All these peaks are observed in the PEKK/HA composite, indicating that it contains HA.

### 3.2. X-ray Photoelectron Spectroscopy

To determine the content of HA in the PEKK/HA composite, X-ray photoelectron spectroscopy was used to obtain the ratios of carbon and calcium in the composite, allowing indirect calculation of the HA content, and the tested PEKK/HA samples are in powder form. (Figure 2)

The measured carbon/calcium ratio is ~7.027:1, which, when converted into a mass ratio, is ~1.015:1. Thus, the HA content of the composite is 50 wt%, i.e., that of human bone.

### 3.3. SEM, TEM and EDS Mapping

Figure 3 shows the SEM images of PEKK and PEKK/HA bulk samples (1 × 1 cm). Unlike pristine PEKK, PEKK/HA shows clear granular structures on its surface.

Figure 4 shows TEM images of PEKK, HA, and PEKK/HA powder samples. Compared to organic matter (polymers), inorganic substances are much darker due to their crystallization. As a result, the organics can be distinguished from the minerals using TEM.

Figure 5 shows EDS elemental mapping results of PEKK and PEKK/HA bulk samples (1 × 1 cm).

Figure 3A,B shows pure PEKK, while C and D show the PEKK/HA composite. It can be seen from the surface topography in B and D that HA is distributed in PEKK in granular form. Figure 4A,B shows pure PEKK, C-D shows pure HA rod-shaped particles of around 30 nm × 100 nm, while E-F shows the PEKK/HA composite, demonstrating that HA is wrapped by PEKK. In Figure 5, the red dots (A) represent carbon, which is from PEKK, while the purple (B) and yellow (C) dots represent calcium and phosphorus, which is found in HA. Hence, the PEKK/HA composite contains both components in a highly distributed and uniform arrangement.

### 3.4. Effect of HA on the Molecular Weight of PEKK

In the present reaction system, HA particles are used as the starting material to synthesize the PEKK/HA hybrid composite, so the effect of HA on the molecular weight of the PEKK had to be established. The molecular weight of PEKK test samples are in powder state. In Figure 6, the first plot shows the molecular weight of PEKK without HA, and the second is for the composite system after removing HA by HCl.

According to the Mark–Houwink formula, the viscosity average molecular weight of the PEKK sample is ~280,000, whereas that of the PEKK/HA is ~330,000, which represents a similar range of molecular weight.

### 3.5. Mechanical Properties of the Composite

Figure 7 shows the results of indentation modulus and hardness analysis of the composite, which is a cylinder with a diameter of 13.0 mm and a height of 5.0 mm. This test method uses nanoindentation. With the increasing number of dental implants and the requirements of other orthopedic applications, biocompatibility is no longer the only consideration, and increasing attention is being paid to biomechanical compatibility, which is vital for improved performance and service lifetime.

The data in Figure 5 and Table 1 show that the average indentation modulus and hardness for the PEKK/HA composite are ~12.5 ± 2.5 GPa and 0.42 ± 0.09 GPa respectively, and the testing method is nanoindentation. The modulus and harness displacement profile varies from sample to sample due to nanostructural variation, which will be discussed below.

## 4. Discussion

The aim of the present work was to synthesize a new class of polymeric hybrid composite consisting of PEKK and HA through a synthetic pathway rather than mechanical mixing. It was hoped that a modulus and hardness close to those of human bone (14.8–22.8 [36] and ~0.47 GPa [8], respectively) could be achieved. The nano indentation test showed that the indentation modulus of PEKK was increased from 7 GPa [36] to 12.5 ± 2.5 GPa, and the nano hardness was increased from 0.2 GPa [36] to 0.42 ± 0.09 GPa with the addition of HA. The mechanical properties of HA/PEKK were close to those of human bones. The results confirm that these goals were largely achieved with the first use of this new technology. This is because the new synthetic pathway allows very high loadings of nanostructured HA of up to 50 wt% (that of natural bone is between 50 and 70 wt%) in a highly uniform fashion at both the micro- and macroscale. Compared with the metal materials used in the current market as dental implants, the mechanical properties of PEKK/HA composites not only meet the requirements, but also avoid the disadvantages of metal materials [37]. Hence, the PEKK/HA composites possess mechanical properties more similar to human bones, which may indicate that it has better osseointegration ability.

Some studies showed that various clinically reinforced PEEK composites have been developed, such as carbon fiber-reinforced polyetheretherketone (CFR-PEEK) and glass fiber-reinforced polyetheretherketone (GFR-PEEK). The elastic modulus of CFR⁃PEEK can be as high as 18 GPa, and that of GFR-PEEK can be as high as 12 GPa, both of which are relatively close to the elastic modulus of normal human cortical bone [38,39,40]. Compared to PEEK, PEKK has better thermal and dimensional stability. PEKK is easier to mold and process, cheaper to produce, and has a wide range of mechanical properties [40].

Based on the bone-like stiffness of PEKK polymer, it is anticipated that by adding HA in the synthesis process, HA/PEKK composites with an elastic modulus equivalent to natural bone can be obtained without changing the molecular weight of PEKK, and at the same time, the PEKK polymer has biological activity [28]. First, the successful synthesis of PEKK without phase segregation in the formed PEKK/HA hybrid composite had to be confirmed, and the fact that the added HA particles do not affect the molecular weight of PEKK in the composite had to be demonstrated.

Figure 1 shows that the characteristic carbonyl peak of PEKK is observed for both pure PEKK and PEKK/HA. This demonstrates that using HA as a starting material does not affect the formation of the PEKK polymer. Both the infrared peak of HA and the infrared peak of PEKK can be observed in the PEKK/HA complex, indicating that PEKK and HA are contained in PEKK/HA complex. Neither the bands shift nor the new absorption bands are identified, suggesting that the lab-synthesized HA/PEKK composites are the mixture of these two compounds without forming identifiable new chemical bonds.

The weight percentage of HA in the composites was determined based on the results of X-ray photoelectron spectroscopy. Here, the calculation principle is that the carbon in the composite only exists in the PEKK, while the calcium is only found in HA. As a result, the carbon/calcium ratio reflects the content of HA in the composite. X-ray photoelectron spectroscopy revealed a carbon/calcium ratio of 7.027:1 (Figure 2), and when the quantity ratio is changed to a mass ratio, it is ~1:1, the content of HA is ~50 wt% in the composites also changes, as hoped.

The HA particle distribution is another matter of concern. Therefore, SEM and TEM are used to study how HA is distributed in PEKK matrix, and EDS is used to study whether HA is evenly distributed in PEKK matrix without phase separation. It can be seen from the SEM images (Figure 3) that HA is dispersed in PEKK polymer and exists in a granular state. TEM graphs (Figure 4) show that HA particle filler is firmly wrapped in the PEKK matrix, and the interface between HA filler and PEKK matrix is seamlessly and firmly combined. As discussed above, carbon is unique to PEKK, while calcium is unique to HA, so their distributions represent those of PEKK and HA, respectively. Figure 5 shows that carbon and calcium are all well distributed in the composite, demonstrating that the distribution of HA is homogeneous and free from phase segregation down to the micrometer, if not, nanometer scale.

The next question that needed addressing is whether the added HA nanoparticles affect the molecular weight of the PEKK. Accordingly, the molecular weight of the PEKK in the composite was measured after removing the HA with hydrochloric acid. The results shown in Figure 4 confirm that the added HA has no negative effect on the molecular weight of the polymer matrix within a permissible margin of error.

Preparing a material with suitable mechanical properties was the major goal of this work. As shown in Figure 5, the nanoindentation results revealed that the indentation modulus and hardness of point E are 8.1 GPa and 0.29 GPa, respectively, which are very different from other data values. The reason is that the HA content in the composite material at this point is far less than that at other places, resulting in the indentation modulus and hardness values being far less than those at other places. This test result also revealed that adding HA will improve the mechanical properties of PEKK/HA composites. Therefore, it is necessary to test five points for one sample to avoid the dispersion and contingency of sample results. It can be concluded from Figure 5 that the indentation modulus of the composite is 12.5 ± 2.5 GPa, and the hardness is 0.42 ± 0.09 GPa, both of which are similar to natural human bone.

Furthermore, the data in Table 1 and Figure 5 show that the indentation modulus and hardness vary with the depth of the indenter. To explain this phenomenon, two models of the sample were constructed, as shown in Figure 8.

According to the density of HA and PEKK, the distance between HA particles can be calculated considering two ideal conditions:(4)$$\frac{S_{PEKK}}{S_{HA}}=\frac{V_{PEKK}}{V_{HA}}=\frac{\frac{M_{PEKK}}{\rho_{PEKK}}}{\frac{M_{HA}}{\rho_{HA}}}=\frac{\rho_{HA}}{\rho_{PEKK}}=\frac{3.10}{1.29}$$

H1: (5)$$\frac{H1\times L}{R\times L}=\frac{S_{PEKK}}{S_{HA}}$$

H2: (6)$$\frac{\left(H2+R\right)\times R-\frac{\pi }{4}R^{2}}{\frac{\pi }{4}R^{2}}=\frac{S_{PEKK}}{S_{HA}}$$

Case H1: If all the HA rods are parallel to each other and to the surface of the sample, the interparticle distance H1 = 144.2 nm

Case H2: If all the HA rods are parallel to each other but perpendicular to the surface of the sample, the interparticle distance H2 = 100.4 nm.

In reality, the interparticle distances will vary from these ideal situations to a great extent, with some being much higher and some much lower. HA is also randomly distributed in the matrix in all directions. Accordingly, the real distance will be intermediate between the two ideals, i.e., somewhere between 100.4 and 144.2 nm.

The testing indenter is much shorter than the interparticle distance. As the indenter penetrates from the surface of the testing sample into the bulk of the sample, the measured hardness and indentation modulus will not be constant, instead depending on the local structure around the indenter. So, the indentation modulus and hardness of the sample will vary from sample to sample and from distance to distance, as shown in Figure 5.

The macro-mechanical properties of the composite are reflected by its nanoindentation load and displacement curves. Based on the results, its yield strength was estimated. The tested sections of the sample were examined by SEM, revealing highly deformed areas containing elongated PEKK polymer strands, imparting excellent fracture toughness. Thus, the newly developed hybrid is expected to be much tougher than natural bone.

The above data confirm that the newly developed PEKK/HA hybrid composite exhibits good biomechanical compatibility and thus the potential of being used for dental implants and other orthopedics. Furthermore, the established biocompatibility of PEKK and bioactivity of HA indicate that the hybrid composite will be a good material for dental implants owing to its biomechanical and biological compatibility [28].

## 5. Conclusions

Here, a new synthetic method for the production of a new class of polymeric inorganic hybrid biomaterials with potential for dental implant applications and other orthopedic applications owing to their excellent mechanical properties and biomechanical compatibility was developed. The new hybrid biomaterial is a composite consisting of PEKK and HA. Nanoparticulate HA was successfully employed as a starting material to load PEKK, and the high HA loading of 50 wt%, which is close to that of natural human bone, was achieved. Excellent homogeneity of HA distribution in the PEKK matrix was also achieved without phase separation. Furthermore, the mechanical properties of the matrix are comparable to those of human bone. The new hybrid composite has a higher yield strength than that of natural bone as well as good ductility, making it tougher than natural bone. Therefore, the new PEKK/HA biomaterial may serve as a base for a range of dental applications and oral implantology.

Although the PEKK/HA composite exhibits excellent mechanical matching and biocompatibility with biological bone, there are still some issues to be addressed for its application to human bone grafting, which are mainly concerned with the dispersion uniformity of HA in the PEKK/HA complex; the difficulty of processing PEKK/HA into biological bone; and the toxicology of PEKK/HA composite when introduced into a biological skeleton.

## Figures and Tables

**Figure 1 jfb-13-00220-f001:**
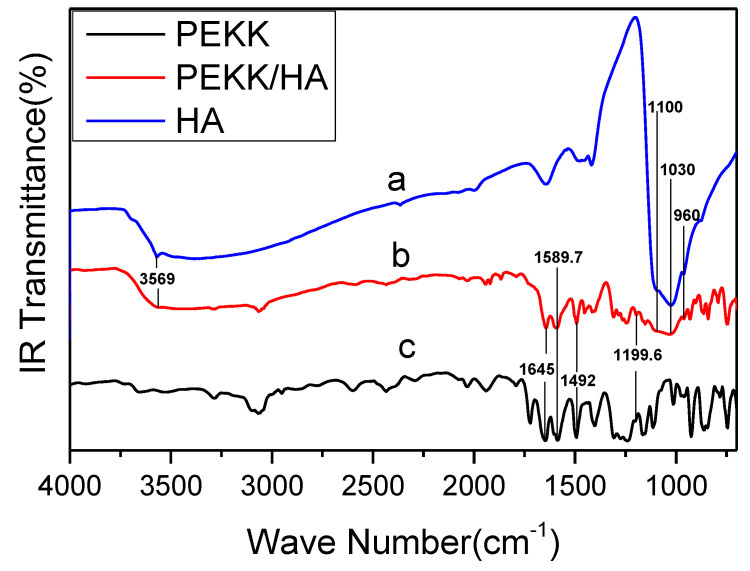
Back reflection IR spectra of (a) HA, (b) the PEKK/HA composite, and (c) PEKK.

**Figure 2 jfb-13-00220-f002:**
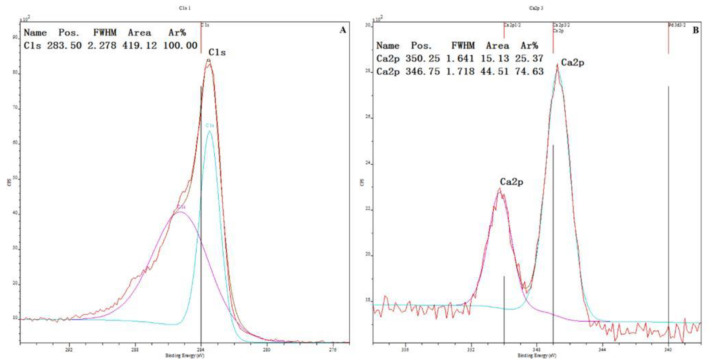
X-ray photoelectron spectroscopy results for the PEKK/HA composite. (**A**): Carbon from PEKK; (**B**): calcium from HA [34].

**Figure 3 jfb-13-00220-f003:**
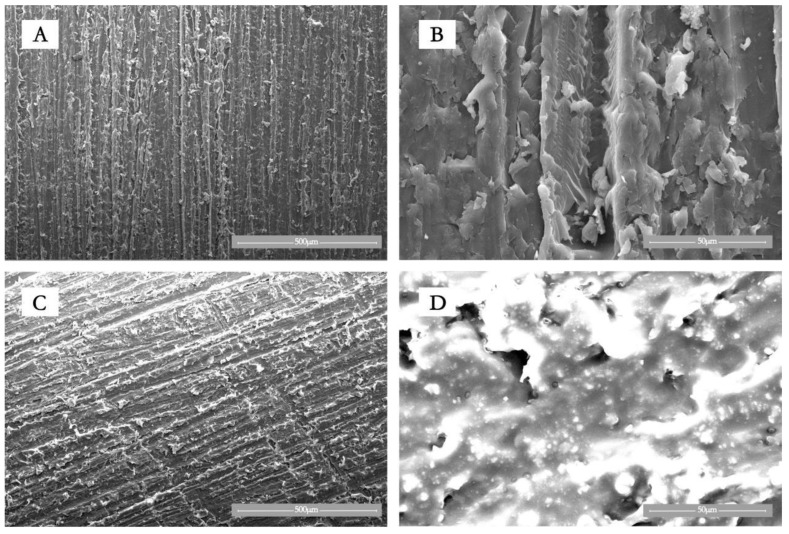
SEM images. (**A**,**B**): Pure PEKK. (**C**,**D**): PEKK/HA composite [35].

**Figure 4 jfb-13-00220-f004:**
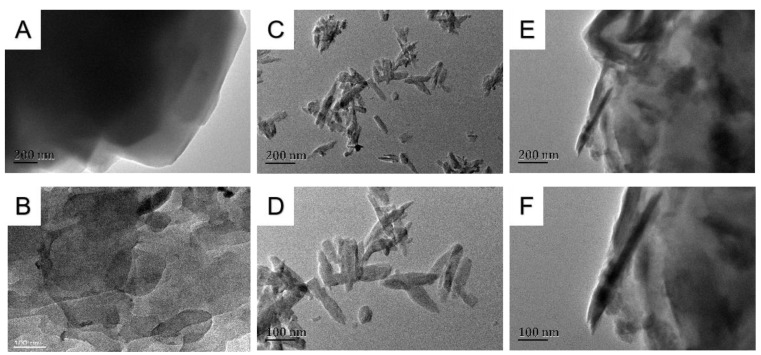
TEM images. (**A**,**B**): Pure PEKK. (**C**,**D**): Pure HA. (**E**,**F**): PEKK/HA composite.

**Figure 5 jfb-13-00220-f005:**
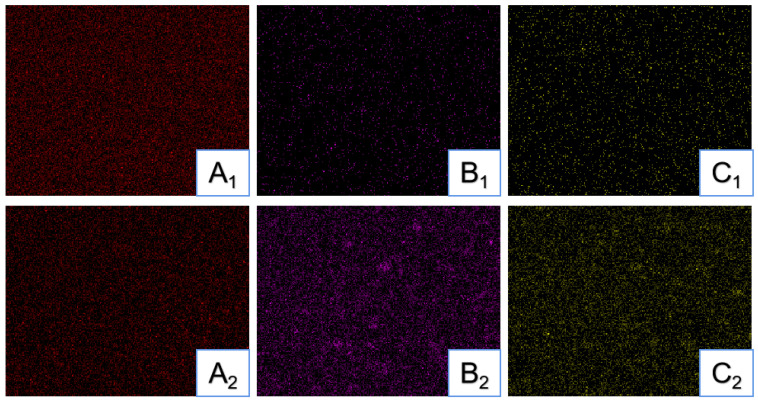
EDS elemental mapping results showing the distributions of (**A_1_**) carbon in PEKK; (**A_2_**) carbon in PEKK/HA; (**B_1_**) calcium in PEKK; (**B_2_**) calcium in PEKK/HA; (**C_1_**) phosphorus in PEKK; and (**C_2_**) phosphorus in PEKK/HA. All images are 10 μm × 10 μm.

**Figure 6 jfb-13-00220-f006:**
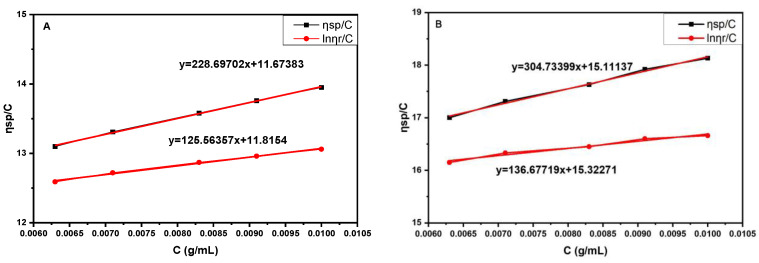
Plots of η_sp_ and lnη_r_/C against C to determine the viscosity average molecular weight of PEKK. (**A**) PEKK without HA. (**B**) PEKK/HA composite.

**Figure 7 jfb-13-00220-f007:**
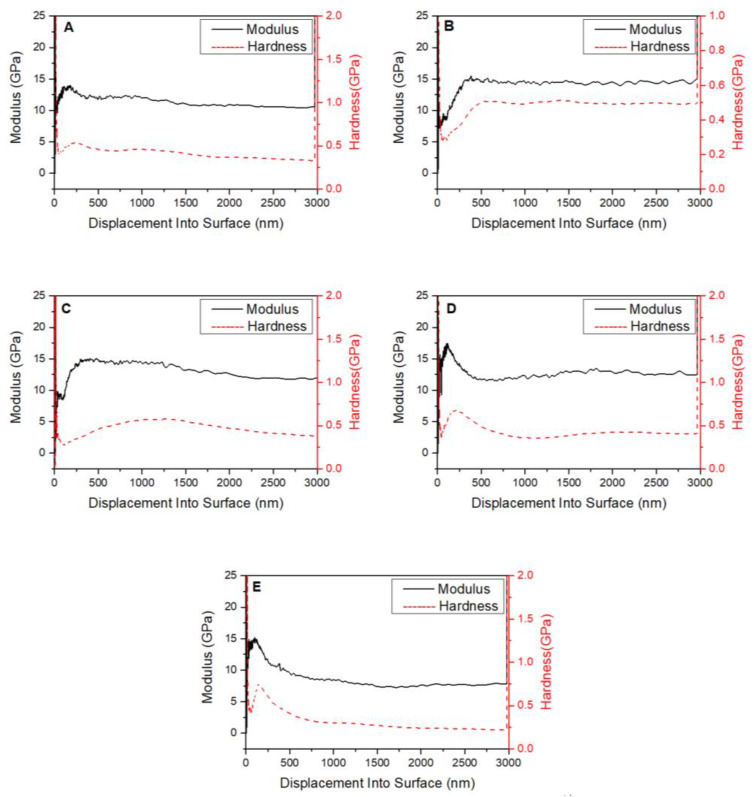
Indentation modulus and hardness results for the PEKK/HA composite sample after thermo-compression molding and polishing. (**A**–**E**) represents the indentation modulus and hardness results of five test points of PEKK/HA composite samples after thermo-compression molding and polishing.

**Figure 8 jfb-13-00220-f008:**
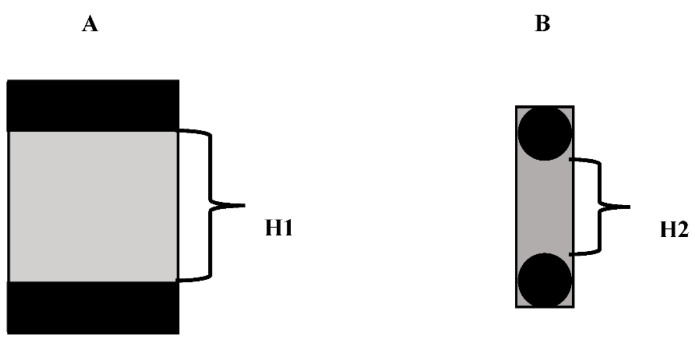
Models of HA (black) wrapped in PEKK (grey). (**A**) All the HA rods parallel to each other and to the surface of the sample. (**B**) All the HA rods parallel to each other and perpendicular to the surface of the sample.

**Table 1 jfb-13-00220-t001:** Average indentation modulus and hardness values for five different points (A–E) on the same sample.

Test	Average Indentation Modulus (GPa)	Average Hardness (GPa)
A	11.5	0.42
B	14.5	0.5
C	13.8	0.51
D	12.4	0.4
E	8.1	0.29
Mean	12.1	0.42
Std. Dev.	2.5	0.09
% COV	20.61	20.99

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
