# Peer review of "Preparation and Characterization of a Polyetherketoneketone/Hydroxyapatite Hybrid for Dental Applications"

_jfb, 2022, doi:10.3390/jfb13040220_

Round 1

Reviewer 1 Report (Previous Reviewer 1)

This is  a nice topic and very interesting from mechanical properties. The osseointegration and its osteogenic potency is named as excellence. How do you come to such conclusion? Here biological investigations are missing so this aspect is really far fetched and should be mentioned with caution.

Moreover how do you define "excellent" mechanical behaviour as long term stability which cause most of the probelms in dental implants today are not yet made ?? We have a lot of nice materials which fail in long term testing and thus are not used in the oral cavity.

Author Response

Reviewer 2 Report (Previous Reviewer 3)

Title: good

  Abstract: good   Keywords: good   Introduction: - Please follow MDPI style for the citing of reference in the text - Line 57: moduli? - Please clarify the aim of the present study at the last paragraph of the intro and provide a null hypothesis   Methods: - good structure but please indicate the sample size and the statistical analysis tests and software   Results: - please use Figure 3 and Figure 4 for Figure 3-2 and Figure 3-2 - please provide the same magnification in Figure 3 to compare PeKK and PEKK/HA - More TEM images for PEKK/HA are recommended - Table 1: what are A, B, C.... and where is the statistical test   Discussion: Until now, there is no real discussion associated with a comparison to other similar previous articles No references are cited in the discussion!!!! I can not accept the article if the authors don't show a real discussion of their results    References: - Please update the references, we are in 2022 - Various errors are presented in the reference list

Round 2

Reviewer 2 Report (Previous Reviewer 3)

Dear Authors,

I asked for SEM images with similar magnifications to compare the groups

It is important to use the same magnification !!!

Please provide it

Best regards

Author Response

This manuscript is a resubmission of an earlier submission. The following is a list of the peer review reports and author responses from that submission.

Round 1

Reviewer 1 Report

the article is not understandable in its current form. The citations are not up to date. Please do a major revision for this very interesting topic prior to sending it to journal.

Thanks

Reviewer 2 Report

Major flaws

1. Title should be revised - please avoid short forms.

2. Number of samples tested? Power sample size missing?

3. In comparison with nature human bone part missing?

4. Figure 4 - 50 micrometer only? Better to visualize with more power. Current figures are of poor quality.

5. Figure 5 - figures are of poor quality.

6. Abstract: in a structured way of planning - lots of vital information details are missing.

Reviewer 3 Report

Title: please don't use abbreviations

Abstract: Please verify the typing of PEKK   Keywords: Same comment for the abstract   Introduction: - L 36: Please use the complete name for PEKK - L 36: FDA - References following MDPI style - L56-57: need REF - L 75: "PEEK" same comment for PEKK in the abstract - L 90-92: need REF - L 93-96: need REF - L 108-112: need REF -L 113: HA first appearance - The aim of the study at the end of the intro should be clarified   Materials & methods: -2.1.materials: please indicate the city, country of each company - L141: AlCl3: please correct the 3 - L 151: what is the icy water? - L 166: spraying gold? it's not sputter-coating with a machine? - L 167: "to get the information of the distribution" normally we use EDX not the SEM - L185: the conditions of thermo-compression modeling and the polishing? - No statistical analysis? - Any sample size test?   Results: - L 198: please correct the -1 in 1645cm (and there are lots of -1 to be corrected) - Where are the images of TEM, Figure 3 only SEM images. The legend of Figure 3 is not correct. I can't understand the results of this part. Please clarify.  - 3.4.: in the M&M section, there is no information on the MAPS technology - In the TAble 1: no statistical analysis? How many samples?   Discussion: No comparison with other similar studies? no references? The limitations?

Round 2

Reviewer 3 Report

good responses